# A Millimeter-Wave Broadband Multi-Mode Substrate-Integrated Gap Waveguide Traveling-Wave Antenna with Orbit Angular Momentum

**DOI:** 10.3390/s24041184

**Published:** 2024-02-11

**Authors:** Qiu-Hua Lin, Da Hou, Lihui Wang, Pengpeng Chen, Zhiyong Luo

**Affiliations:** 1School of Electronics and Communication Engineering, Sun Yat-sen University, Shenzhen 518107, China; linqh26@mail2.sysu.edu.cn (Q.-H.L.); houd5@mail2.sysu.edu.cn (D.H.); wanglh65@mail2.sysu.edu.cn (L.W.); chenpp27@mail2.sysu.edu.cn (P.C.); 2Pengcheng Laboratory, Shenzhen 518107, China

**Keywords:** orbit angular momentum (OAM), broadband multi-mode antenna, substrate-integrated gap waveguide (SIGW), millimeter-wave, phase difference

## Abstract

Orbit angular momentum (OAM) has been considered a new dimension for improving channel capacity in recent years. In this paper, a millimeter-wave broadband multi-mode waveguide traveling-wave antenna with OAM is proposed by innovatively utilizing the transmitted electromagnetic waves (EMWs) characteristic of substrate-integrated gap waveguides (SIGWs) to introduce phase delay, resulting in coupling to the radiate units with a phase jump. Nine “L”-shaped slot radiate elements are cut in a circular order at a certain angle on the SIGW to generate spin angular momentum (SAM) and OAM. To generate more OAM modes and match the antenna, four “Π”-shaped slot radiate units with a 90° relationship to each other are designed in this circular array. The simulation results show that the antenna operates at 28 GHz, with a −10 dB fractional bandwidth (FBW) = 35.7%, ranging from 25.50 to 35.85 GHz and a VSWR ≤ 1.5 dB from 28.60 to 32.0 GHz and 28.60 to 32.0 GHz. The antenna radiates a linear polarization (LP) mode with a gain of 9.3 dBi at 34.0~37.2 GHz, a *l* = 2 SAM–OAM (i.e., circular polarization OAM (CP–OAM)) mode with 8.04 dBi at 25.90~28.08 GHz, a *l* = 1 and *l* = 2 hybrid OAM mode with 5.7 dBi at 28.08~29.67 GHz, a SAM (i.e., left/right hand circular polarization (L/RHCP) mode with 4.6 dBi at 29.67~30.41 GHz, and a LP mode at 30.41~35.85 GHz. In addition, the waveguide transmits energy with a bandwidth ranging from 26.10 to 38.46 GHz. Within the in-band, only a quasi-TEM mode is transmitted with an energy transmission loss |$S_{21}$| ≤ 2 dB.

## 1. Introduction

Electromagnetic waves (EMWs) possess both a linear polarization (LP) mode and angular momentum (AM), with the latter being further divided into spin angular momentum (SAM) and orbital angular momentum (OAM) [1]. SAM includes left/right-hand circular polarization (L/RHCP) modes. Since 2012, OAM has received widespread attention due to its potential to increase communication capacity [2,3,4], including research on its broadband, OAM multi-mode/angular momentum hybrid mode, and high-gain performance. According to [2,3,4], it is not difficult to find that antennas carrying OAM have always been a key research topic in 5G technology, and even Chinese experts have included it in the 6G development plan [3]. Therefore, OAM is of great significance in addressing the high-capacity and high-data transmission rate requirements of 5G millimeter wave communication.

The generation mechanism of OAM is that the phase of EMWs changes continuously in integer periods of 2π. To generate the OAM mode, various types of antenna have been developed. This article is divided into four categories [4]: (1) single microstrip patch antennas, (2) array antennas, (3) meta-surface/material (meta-sur.) antennas, and (4) traveling-wave antennas. The low-cost and simple single-microstrip patch antenna generates OAM by designing the number and location of the feed points, changing the antenna structure, and allowing the current to produce at least 360° phase changes on the antenna element. This results in an OAM with a fractional bandwidth (FBW) of 1.82% and not more than two modes [5]. It is not difficult to see that this type of antenna has a drawback of having a narrow bandwidth and low gain [5,6].

Array antennas can also generate different modes of OAM by controlling the phase change. Currently, various phased array antennas have been proposed to generate OAM, such as patch array [7], circular ring phased array [8], and microstrip patch phased array [9,10]. According to the mechanism of generating OAM, these antennas are designed with a phase-shifted feed network. Based on the open resonant cavity theory and circular array theory, a high-order Laguerre–Gaussian OAM mode (i.e., $HE_{21}$) with a phase difference of 90° is generated using eight aperture-coupled patch antenna elements [7]. To obtain multiple modes, [9] designs a reconfigurable antenna with four phase delay lines with a phase difference of N × 2π/M, where N and M are the phase order and the array element number along period direction, respectively. To simplify the phase-shifted feed network of phased array antennas, [8] uses a “+”-shaped feed network to feed the circular array elements with the same phase in the array.

Another simplified feed network and increased bandwidth and gain scheme is to use meta-surface/material technology to generate OAM [11,12,13]. Meta-surface/material can achieve sudden changes in the amplitude or phase of EMWs by changing the shape and size of resonant elements. To enhance the communication range, a lens-integrated meta-material structure (MMS) has been proposed to generate two orbital angular momentum (OAM) modes with a high-directivity broadband characteristic [11]. In [13], using an antenna with a high gain of 10.1 dBi as a feed source, the authors propose an OAM antenna based on a folded reflective array. The surface can generate *l* = 0, *l* = +1, and *l* = +2 OAM mode reconfigurable beams with gains of 16.2 dBi and 13.9 dBi, respectively. The above meta-surface OAM antenna has significant effects in improving gain and simplifying the feed network. However, the profile height of the antenna and the poor stability of the antenna due to the separation of the polarization plate and the antenna (with a focal length of generally 5λ) cannot be ignored.

OAM antennas are numerous, and but they can be classified into two categories: scalar beam generation systems and vector beam generation systems. The former works by introducing additional spatial phase factors $e^{jl\pi }$ to traditional scalar antenna systems, as described in [5,6,7,8,9,10,11,12,13]. The latter functions by combining some cylindrical vector modes [14,15,16] to generate non-zero OAM topological charges, as shown in [17]. Traveling-wave OAM antennas can be classified into vector beam generation systems, which can be divided into two categories: spiral phase plates (SPP) [14], circular waveguides [15], and Archimedean spiral arms [16]. In [14], a millimeter-wave ultra-low-reflectivity SPP OAM was proposed. However, SPP systems usually can only generate a single OAM beam. [16] proposes a four-arm planar spiral OAM antenna for generating *l* = 0, −1, −2, and −3 OAM modes. By adjusting the branches between the feed port and the corresponding spiral arm, the antenna can easily achieve good matching. At 5.8 GHz, the FBW of the antenna is 6.89%. These OAM antennas [14,16] have a low profile and lightweight characteristics and theoretically generate an infinite number of OAM modes. However, this type of antenna still has the disadvantages of narrow bandwidth and low gain.

Waveguides are often used to design antennas with a high-frequency band, high gain, and a wide bandwidth [17,18,19]. In [18], a gap waveguide (GW) slot antenna with longitudinal slots cut on the top broad wall is presented. Its main feature is that the antenna radiates LP mode within the operating frequency band, while suppressing the propagation of electromagnetic waves outside of this band. In [19], a substrate-integrated waveguide (SIW) millimeter wave 2 × 2 broadband CP-OAM antenna array is proposed, whose elements consist of two layers of meta-surface patch. The antenna utilizes a sequential rotation feeding method to simultaneously generate dual OAM modes. The results indicate that the impedance bandwidth ($S_{11}$ ≤ −10 dB) and 3 dB axis ratio (AR) is 32.3% (from 26 to 36 GHz). Focusing on 5G millimeter wave technology, the substrate-integrated gap waveguide (SIGW) [20] technology emerged. SIGW has achieved certain results in antenna design, such as a electromagnetic dipole antennas with a low profile and high gain [21], 60 GHz broadband circularly polarized slot antennas [22], a circularly polarized patch array antenna [23], and a 25 GHz broadband filtering antenna [24]. However, the above research treats the SIGW as a single-port feed structure of an antenna, and the EMWs radiated by the antenna are LM and SAM, such as line polarization (LP) beam, and left/right-hand circular polarization (L/RHCP), without introducing multiple modes of OAM. The perfect magnetic conductor (PMC) structure inherent in the SIGW can specifically address the current problems of the large divergence angle and low gain of OAM antennas. More importantly, the broadband filtering characteristics of the waveguide can effectively solve the current problem of narrow bandwidth of OAM antennas.

This paper proposes a millimeter-wave broadband multi-mode SIGW traveling-wave antenna, as shown in Figure 2. Following the mechanism of generating the OAM mode, the conductor of SIGW is innovatively utilized to introduce phase delay, exciting the “L”-shaped radiating elements etched on the waveguide with continuously varying phases. The antenna operates at 28 GHz, with a −10 dB bandwidth ranging from 25.50 to 35.55 GHz and a VSWR ≤ 1.5 dB from 28.60 to 32.0 GHz and 28.60 to 32.0 GHz. The antenna radiates a LP mode with a peak gain of 9.3 dBi at 25.50~25.90 GHz, a *l* = 2 SAM–OAM (i.e., CP–OAM) mode with a peak gain of 8.04 dBi at 25.90~28.08 GHz, a *l* = 1 and *l* = 2 hybrid OAM modes with a peak gain of 5.7 dBi at 28.08~29.67 GHz, a SAM (i.e., L/RHCP) mode with a peak gain of 4.6 dBi at 29.67~30.41 GHz, and a LP mode at 30.41~35.55 GHz. In addition, the waveguide transmits energy with a bandwidth ranging from 26.10 to 38.46 GHz. Within the in-band, only a quasi-TEM mode is transmitted with an insertion loss |$S_{21}$| ≤ 2 dB. The structure of this paper is as follows: II: design and analysis; III: simulation results; and IV: providing a summary of this work.

## 2. Design and Analysis

As shown in Figure 1, to meet the growing demand for data transmission, multiplexing techniques based on amplitude, frequency, and polarization are commonly used to improve information transmission [4]. OAM is the angular momentum of electromagnetic waves, which theoretically has an infinite number of orthogonal modes. Unlike traditional wireless communication technologies, the mode multiplexing principle of OAM systems utilizes the orthogonality between different modes of OAM beams, and uses the mode order *l* of OAM waves as a modulation parameter [2]. Linear momentum transmission of information is simple, robust, and easy to implement, but due to the unimodal linear momentum of EM waves, there is only one independent transmission channel at each carrier frequency, resulting in spectrum waste. Therefore, for 5G communication, the design technology of orbital angular momentum antennas with more mode order *l* is of great significance.

### 2.1. Configuration of the Proposed Antenna

The overall view of the antenna is shown in Figure 2, which consists of a SIGW and slot radiating elements etched on the waveguide. The slot elements consists of two groups: the “L”-shaped elements, i.e., #1-0~#1-8, and the “Π” shaped elements, i.e., #2-0~#2-3. The “L”-shaped group is arranged in a ϕ = 40° circular order around the *z*-axis. The “Π”-shaped group is also arranged around the *z*-axis at 90°, but the two elements are orthogonal (90°) to each other. The design parameters are shown in Table 1.

The SIGW consists of two layers of dielectric plates (Substrate 1 and 2). Substrate 1 (Rogers RT5880, $\varepsilon_{r}$ = 2.2, δ = 0.0009) is the gap layer dielectric of the SIGW, with a metal layer printed on its upper surface. There are two rows of periodic metalized mushroom-shaped vias along the *x*- and *y*-direction on Substrate 2 (Rogers RO4003, $\varepsilon_{r}$ = 3.38, δ = 0.0027), respectively, with a metal layer printed on its lower surface. A feed structure is designed at the in- and output ports. The feed structure consists of two coplanar segments of a 50 Ω microstrip line and a conductor of SIGW. Its design parameters are shown in Table 2.

### 2.2. Working Mechanism

The equivalent circuit technique is applicable to the working mechanism of antennas, including the resonant characteristics, impedance characteristics, and propagation characteristics. According to the proposed antenna shown in Figure 2, this paper studies its equivalent transmission line circuit model, as shown in Figure 3. In Figure 3, there are three parts: the first part is the equivalent transmission model of the dual-port SIGW, whose characteristic impedance $Z_{SIGW}$ and propagation constant $\beta_{SIGW}$ have been defined in [23]. To better match and measure the antenna, this paper delays the length of the dielectric plate, and its corresponding characteristic impedance and propagation constant can be defined by a microstrip line (MSL) that is equal to $Z_{MSL}$ and is equal to $\beta_{MSL}$, respectively. The second part is the “L”-shaped group of slot elements, which shows parallel relationships and dielectric losses. Therefore, its resonant characteristics of the antenna are characterized by RLC resonance, with a total resistance of $R_{1}$, a total inductance of $L_{1}$, and a total capacitance of $C_{1}$, which are coupled with the waveguide at a 1:m energy ratio and are thus excited. The third part is the “Π”-shaped group with the etched slot element. There is also a parallel relationship and a dielectric loss due to the presence of four “Π”-shaped slots around the *z*-axis at 90° to each other. Therefore, its resonant characteristics are characterized by RLC resonance, with corresponding total resistance $R_{2}$, total inductance $L_{2}$, and total capacitance $C_{2}$, which are coupled with the waveguide in a 1:n energy ratio.

The studies in [1,2] have shown that the generation mechanism of OAM is that the phase of EMWs continuously changes over integer periods of 2π. In [23], a circuit model of phase change and conductor length has been obtained, and a 2 × 2 CP array antenna with a sequential rotating feed network with 90° phase change has been designed. In this paper, as shown in Figure 4, the conductor of SIGW is used to introduce a traveling wave. When EMW is transmitted from the input to the output port, there is a phase change due to the different lengths of the conductor, as shown in Figure 5. Therefore, it will cause different phase differences in the radiating elements when the electromagnetic waves propagate and couple to the slot elements.

We know that *k* = 2π/λ, where *k* is the wave number of EMWs. When *k* takes the value of 1, EMWs have a wavelength of 2π. In Figure 4, because all the slot elements are excited by the same in- and output ports, and the conductors are of equal width, the two sets of slot elements are excited with equal amplitude power. Assuming that the initial phase of the #1-0 slot element in the “L”-shaped group is $\varphi_{1}$, the phase of the #1-1 slot element is $\varphi_{1}$ + Δ$\varphi_{11}$. Therefore, we can obtain the excitation phase of the #1-8 slot element as follows: $\varphi_{1}$ + Δ$\varphi_{11}$ + Δ$\varphi_{12}$ + Δ$\varphi_{13}$ + Δ$\varphi_{14}$ + Δ$\varphi_{15}$ + Δ$\varphi_{16}$ + Δ$\varphi_{17}$ + Δ$\varphi_{18}$≈$\varphi_{1}$ + 2π [2]. At this point, when EMWs are transmitted from the input to the output port, the slot radiating elements in the “L”-shaped group are excited and generate OAM modes with a mode rank *l* ≤ 2 [2]. However, for the “Π”-shaped group, due to the 90° relationship between its elements, there is no phase shift added by the conductor between the #2-0 and #2-1, and #2-2 and #2-3 slots. There is a phase shift Δ$\varphi_{23}$ added by the conductor between the #2-0 and #2-2, and #2-1 and #2-3 slots. Assuming that the initial phase of the #2-0 slot element is $\varphi_{2}$, then the phase of the #2-1 slot element is $\varphi_{2}$ + 90°, and the phase of the #2-2 slot element is $\varphi_{2}$ + 90° + Δ$\varphi_{23}$.

Finally, the excitation phase of the #2-3 slot element is: $\varphi_{2}$ + 90° + Δ$\varphi_{23}$ + 90° ≈ $\varphi_{2}$ + 2π. When EMWs are transmitted from the input to the output port, the slot radiating elements in the “Π”-shaped group are excited and have modes with a mode rank *l* ≤ 1 [2]. Because there is a 90° relationship between the elements, it is easy to obtain a SAM (i.e., circularly polarized mode) mode [23].

In our previous research work [25], we have analyzed the working mechanism of SIGW forming electromagnetic band-gap (EBG) in detail and verified its application in filters [26] and filtering antennas [23]. Now we will briefly analyze it.

The SIGW is a substrate-integrated achievement of the ridge gap waveguide (RGW), featuring integration and lightweight structure, and exhibiting a EBG, broadband, and low insertion loss (IL) characteristics. Firstly, in fact, SIGW is a waveguide formed by packing a microstrip line with a layer of perfect magnetic conductor (PMC) meta-material composed of mushroom-shaped vias. In this paper, we focus on analyzing the role of mushroom-shaped vias in providing waveguides with frequency-selective function and solving dispersion problems. The PMC structure exhibits a high impedance surface within the operating frequency band [27], which has the function of suppressing surface waves and solving the high-frequency dispersion problem of microstrip lines. The SIGW shows one-dimensional periodic characteristics along its conductor, and mushroom-shaped vias consists of the one-dimensional periodic structure, as shown in the Figure 6. The one-dimensional periodic structure is modeled and solved for eigenvalues using CST, resulting in an EBG and only a quasi-TEM in the operating frequency band, as shown in Figure 6.

As can be seen from the above Figure 6, when the conductors are not printed, the mushroom-shaped via shows a two-dimensional periodicity characteristic with an EBG range of 20.50~41.25 GHz. However, when the conductor is printed, the SIGW still exhibits an EBG range of 20.51~39.85 GHz, but only a quasi-TEM propagates along the conductor within the operating band. Therefore, in this paper, the propagation constant (β/m) of the quasi-TEM exhibits a linear relationship with frequency which does not cause a dispersion problem. However, in dispersive media, it no longer linearly related to the frequency, resulting in a dispersion phenomenon.

To better show the changes in the electric field on the conductor and mushroom-shaped vias while the waveguide operates in-band and out-of-band, Figure 7 explains this briefly. From Figure 7, it can be seen that the mushroom-shaped vias have a weak electric field at all bands. However, there is stronger energy transmission from the input port to the output port, and a stopband transmission to the output port while operating out-of-band.

Then, the MSL with the same size as the SIGW is simulated to obtain its reflection and transmission coefficients, as shown in Figure 8. The transmission coefficient $S_{21}$, i.e., insertion loss (IL), is equal to 0.3 dB and the reflection coefficient $S_{11}$, i.e., return loss (RL), is >18 dB. However, it can cause electromagnetic interference out-of-band due to the lack of frequency selectivity.

Finally, to explain the effect of mushroom vias on S-parameters, a comparison to the MSL and SIGW is shown in Figure 9.

Based on the 5G millimeter wave bands, which include 21.4~22.0 GHz, 24.25~27.5 GHz, 27.9~28.2 GHz, 31.0~31.3 GHz, and 38.0~39.5 GHz [25,28], announced by the Wires Radiocommunication Conference-19 (WRC-19), the SIGW operates at 21.0~40.0 GHz. The SIGW is modeled and calculated by HFSS.2020, and then its S-parameters are obtained. As shown in Figure 9, the S-parameter values can be seen, showing that the IL is equal to 0.8 dB and RL is >20 dB at 25.5~34.5 GHz, which is similar to the MSL, although there is a significant difference ranging from 15.0 to 21.0 GHz and from >40 GHz band. There is significant attenuation and finite transmission zero (FTZ) out-of-band, which enhances the frequency selectivity of the SIGW. However, the MSL does not have transmission suppression and frequency selectivity characteristics out-of-passband, which increases the difficulty of reducing interference between operating frequency bands and out-of-band interference.

## 3. Simulation Results

The proposed antenna is modeled and solved by using the software HFSS.2020. A detailed description of the simulation settings and parameters is shown in Figure 10. After entering HFSS.2020, an “AngleAnt” project tree is created, and antenna structure materials and model interface will appear automatically. The project tree automatically includes 1: Boundaries, 2: Excitations, 3: Optimetrics, 4: Results, and 5: Field Overlayers. The Boundaries module is mainly used to simulate the characteristics of actual materials, such as printed metal layers, which are generally set as perfect electric conductors (PEC). It is worth noting that simulating the actual atmospheric environment requires setting a material as “vacuum”, with a size larger than 0.25λ. The Excitations module simulates the feeding excitation source. The Optimetrics module is mainly used to optimize the antenna structural parameters according to its design specifications. The Results module includes S-parameters, VSWR, gain, direction pattern, and so on. The Field Overlayers module mainly outputs the electric field and surface current vector maps of the antenna. The antenna structure materials and model interface include solid materials (i.e., number 7) and printed metal layer materials (i.e., number 8), as well as a coordinate system (i.e., number 9). In Figure 10, the SMA connector used for actual antenna testing is also shown to obtain a more realistic antenna performance.

Furthermore, in terms of experimental validation, here we will discuss the microwave darkroom testing environment for the antenna, as shown in Figure 11. The main equipment includes a vector network analyzer, a test stand, a probe, the designed antenna, a computer for data post-processing, and necessary cables. It is worth noting that before antenna testing, it is necessary to calibrate the vector network analyzer and test signal channels using calibration components. The test sampling probe and test stand are responsible for collecting near-field signals of the antenna. The vector network analyzer provides the excitation signal and records the sampling signal from the probe. The probe moves and collects test data at sampling points, which are then delivered to the central control and data processing computer loaded with the test data processing software package to output the test results.

The center frequency was solved to be 28 GHz, and the S-parameters, VSWR, axial ratio (AR), gain, and directional pattern of the proposed antenna were obtained. The S-parameters mainly consider a reflection coefficient $S_{11}$ (dB) and a transmission coefficient $S_{21}$ (dB), as shown in Figure 12.

From the above Figure 12, it can be seen that the −10 dB bandwidth is located at 25.50~35.55 GHz, with a VSWR ≤ 1.5 dB ranging from 28.60 to 32.0 GHz and 28.60 to 32.0 GHz, and the energy transmission loss |$S_{21}$| is ≤2 dB ranging from 26.10 to 38.46 GHz.

The antenna radiates CP beams, which requires an equal electric field amplitude and a 90° phase difference. The antenna proposed in this paper mainly consists of a “Π”-shaped group. Based on a 90° relationship between the elements, assuming that the initial phase of the #2-0 slot element is $\varphi_{2}$, the #2-1 slot element is $\varphi_{2}$ + 90°, and the #2-2 slot element is $\varphi_{2}$ + 90° + Δ$\varphi_{23}$, the excitation phase of the #2-3 slot element is $\varphi_{2}$ + 90° + Δ$\varphi_{23}$ + 90° ≈ $\varphi_{2}$ + 2π. This ensures that the radiation has a CP–OAM mode. This paper compares the axial ratio (AR) curves of the antenna without the “Π”-shaped group slots, as shown in Figure 13.

In Figure 13, it can be seen that the antenna in this paper has an AR of ≤3 located at 25.90~28.08 GHz and 29.67~30.41 GHz in two bands. However, while there are no “Π”-shaped elements, the AR > 3 dB at all operating frequency bands, so it will difficult to generate a CP mode. Therefore, the CP–OAM generation idea of this paper can be verified.

In Figure 14, it can be seen that the antenna radiates a LP mode with a peak gain of 9.3 dBi at 25.50~25.90 GHz.

The electric field and current at 28.0 GHz are shown in Figure 15. In Figure 16, a *l* = 2 CP–OAM (i.e., SAM-0OAM) mode with a peak gain of 8.04 dBi at 25.90~28.08 GHz can be seen. Additionally, there is a weak electric field at the center of the phase [2], which is in accordance with the OAM mode with *l* = 2 as described in [1,2].

In Figure 17, firstly, a *l* = 1 OAM mode with a peak gain of 5.5 dBi is shown in Figure 17a,b, which finds a weak electric field at the center of the phase [2]. Then a hybrid OAM mode with *l* = 2 and *l* = 1 and a peak gain of 5.7 dBi at 28.08~29.67 GHz is shown in Figure 17c,d. From Figure 17, we can find that they both have a weak electric field at the center of phase. In addition, there is a CP mode with a gain of 4.6 dBi at 29.67~30.41 GHz. Table 3 compares various antennas from recent years in terms of design technology (i.e., Tech. column, includes microstrip patch (Pat.), microstrip patch array (Pat.-arr.), meta-surface (Meta-sur.), and waveguide), center frequency (i.e., $f_{0}$ column), factional bandwidth (i.e., FBW column), peak gain (i.e., Gain column), Mode numbers (i.e., Modes column), and size of the antennas (i.e., Size column)).

It can be seen from Table 3 that the center frequency $f_{0}$ of the proposed antenna is $f_{0}$ = 28.0 GHz, covering the 5G millimeter wave range of 25.5~27.5 GHz, 27.9~28.2 GHz, 31.0~31.3 GHz with a FBW of 35.7% (25.5~35.50 GHz). This is a higher frequency band compared to the microstrip patch technology [5,8] and a wider bandwidth compared to the design proposed in the references [5,7,8,13,17,19,24]. To achieve different modes and maintain high gain of the antenna in this paper, it is only necessary to design resonating slots in a sequentially rotating angle on the SIGW, without the need for any other complex meta-surface structure. Modes of the proposed antenna (Pro.) show LP and CP modes with *l* = 0, an OAM mode *l* = 1, a CP–OAM mode *l* = 2, and a hybrid OAM mode of *l* = 1 and *l* = 2, which radiates more modes compared to references [11,14,17,19], in that they radiate OAM modes with *l* = −1 and *l* = +1. With regard to the gain of the OAM mode, it is well known that there is a relatively low gain for an OAM mode, especially a higher order OAM mode. In this table, it can be seen that the proposed antenna radiates a *l* = 2 OAM mode at 25.90~28.08 GHz with a gain of 8.04 dBi, which shows a relatively high gain compared to references [6,9,11,22]. In addition, the volume of this antenna is (1.5, 1.4, 0.2) ·$\lambda_{0}^{3}$, where $\lambda_{0}$ is the wave length of the center frequency $f_{0}$; therefore, there is a smaller volume and lighter quality to this antenna compared to references [7,8,9,13,18,19].

## 4. Conclusions

In this paper, a mm-wave broadband multi-mode SIGW traveling-wave antenna with OAM is proposed. The OAM is generated by using the conductor of the SIGW to conduct electromagnetic waves, with the phase varying along the length of the conductor to produce the necessary phase jump for OAM modes. By fusing the EBG characteristics of SIGW, only quasi-TEM is conducted within the operating frequency band, suppressing coupling with other stray harmonic modes and solving the problems of low gain of OAM antennas. The results show that the proposed antenna has a center frequency of 28 GHz, a 3 dB band located at 25.50~35.85 GHz, and a VSWR ≤ 1.5 dB from 28.60 to 32.0 GHz and 28.60 to 32.0 GHz. The antenna radiates an LP mode with a gain of 9.3 dBi at 25.50~25.90 GHz, a *l* = 2 CP-OAM mode with a gain of 8.04 dBi at 25.90~28.08 GHz, a hybrid OAM mode with *l* = 2 and *l* = 1 at 28.08~29.67 GHz, a CP mode with a gain of 4.6 dBi at 29.67~30.41 GHz, and an LP mode at 30.41–35.85 GHz. In addition, the energy transmission loss |$S_{21}$| is ≤2 dB over the broadband of 26.10~38.46 GHz. The OAM antenna proposed in this paper provides a novel solution for the design of high-capacity antennas for 5G millimeter-wave communication systems.

## Figures and Tables

**Figure 1 sensors-24-01184-f001:**
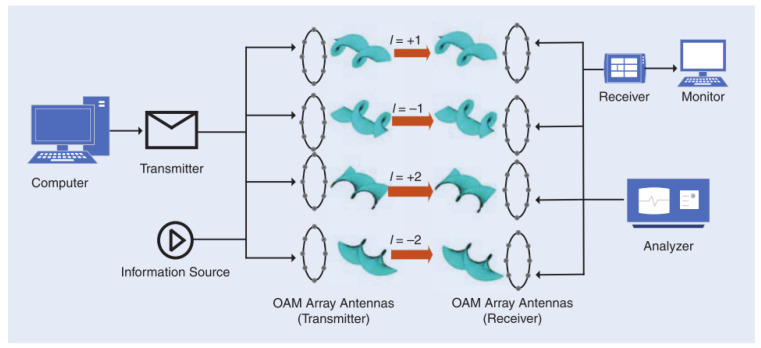
The schematic diagram of OAM beam multiplexing [4].

**Figure 2 sensors-24-01184-f002:**
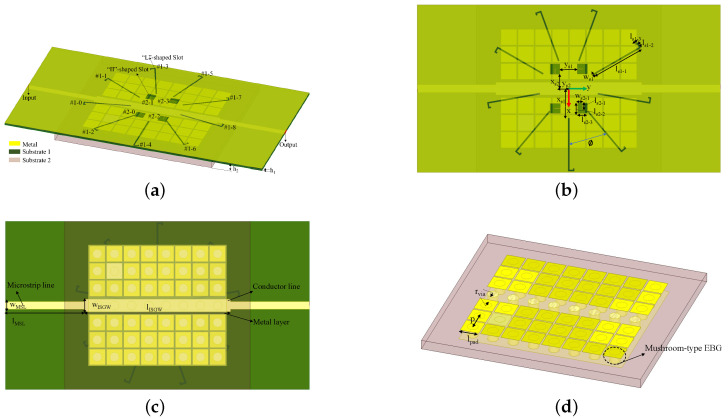
Structure and size of the proposed antenna: (**a**) overall view; (**b**) vertical view; (**c**) bottom view; and (**d**) PMC structure.

**Figure 3 sensors-24-01184-f003:**
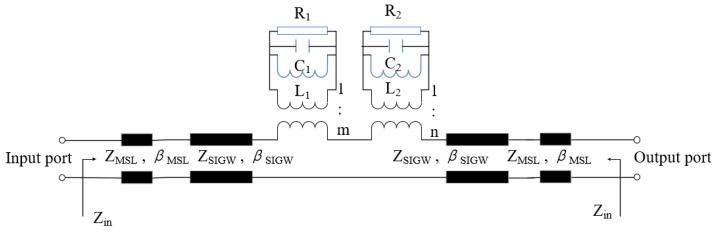
The equivalent transmission line circuit model of the proposed antenna.

**Figure 4 sensors-24-01184-f004:**
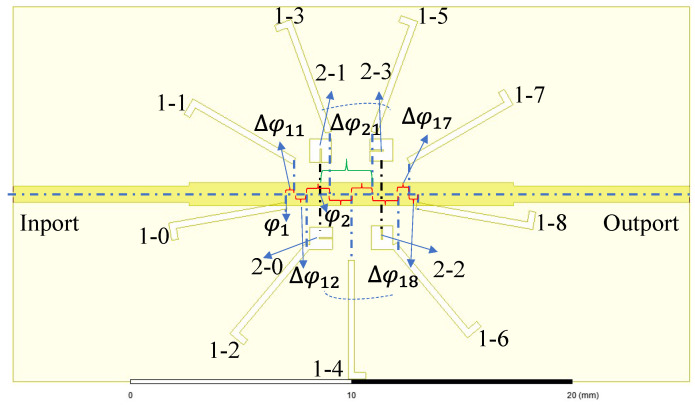
Analysis of radiation working mechanism.

**Figure 5 sensors-24-01184-f005:**
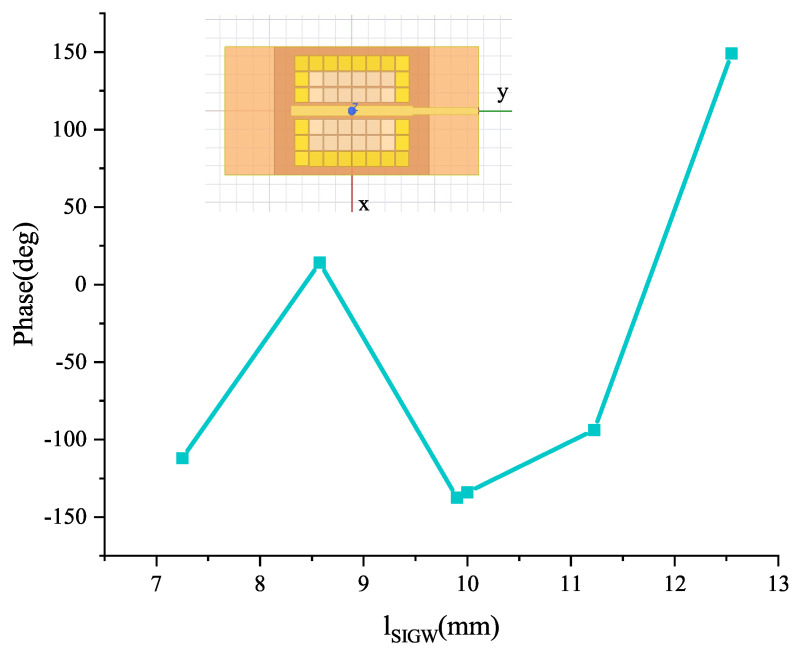
Relationship between $l_{SIGW}$ and phase change.

**Figure 6 sensors-24-01184-f006:**
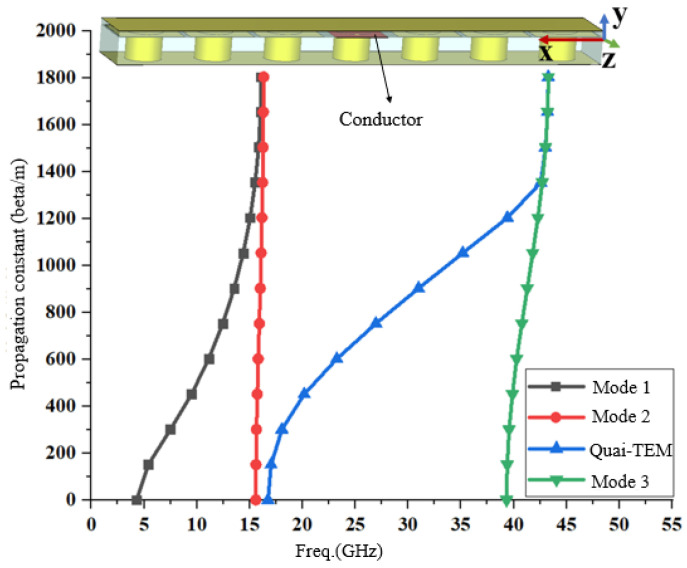
An EBG of the SIGW.

**Figure 7 sensors-24-01184-f007:**
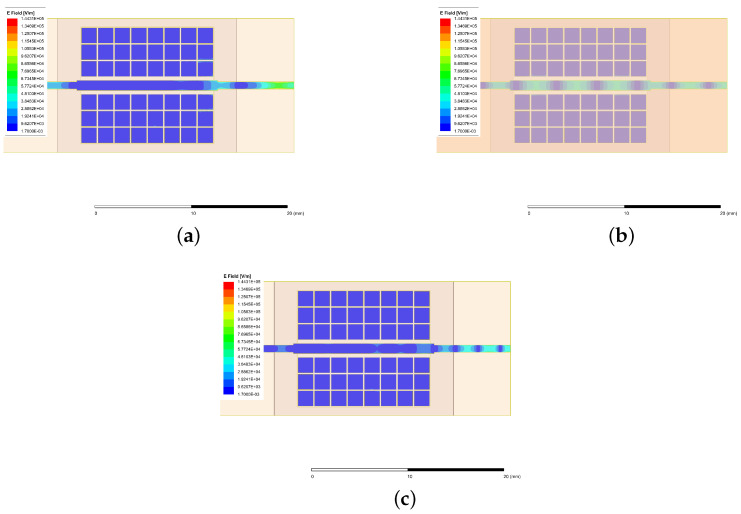
The electric field in-band (at (**b**) 28.0 GHz) and out-of-band (at (**a**) 15.0 GHz and (**c**) 45.0 GHz).

**Figure 8 sensors-24-01184-f008:**
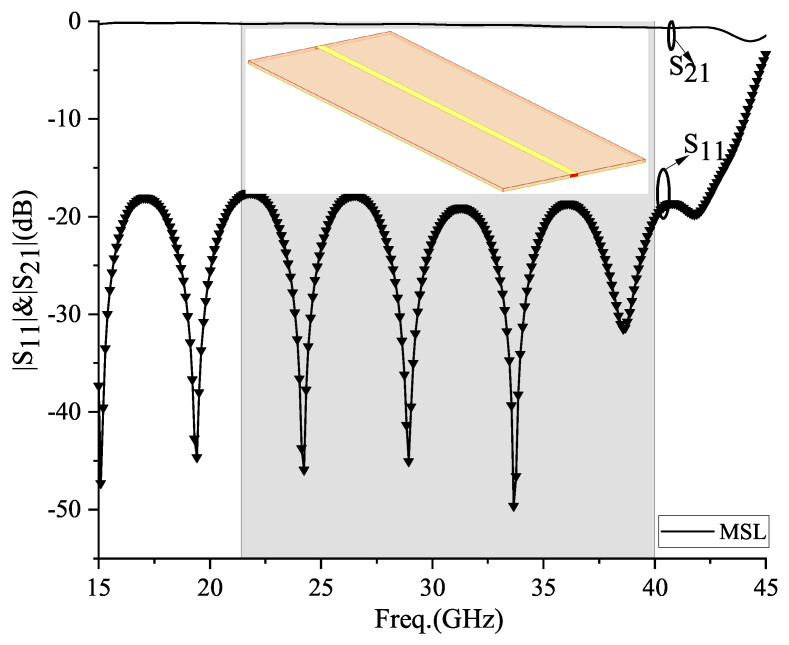
MSL structure and its $S_{11}$ prameters.

**Figure 9 sensors-24-01184-f009:**
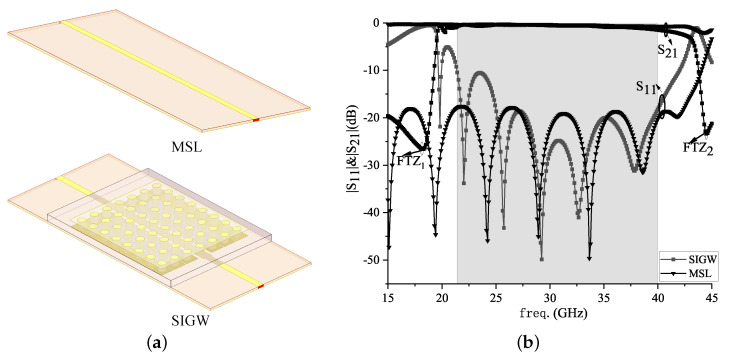
Structures of MSL and SIGW and their |S|-parameters and VSWR: (**a**) Structures; (**b**) |S|-parameters.

**Figure 10 sensors-24-01184-f010:**
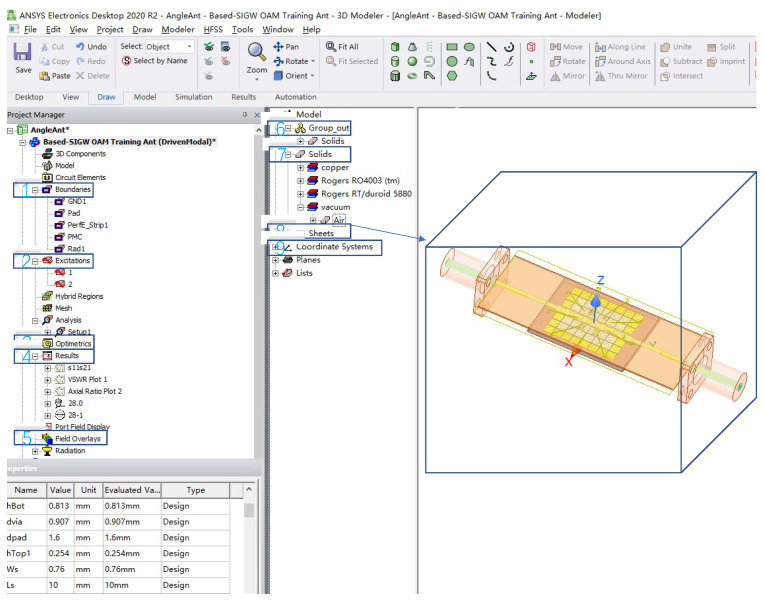
Simulated interface by HFSS.2020.

**Figure 11 sensors-24-01184-f011:**
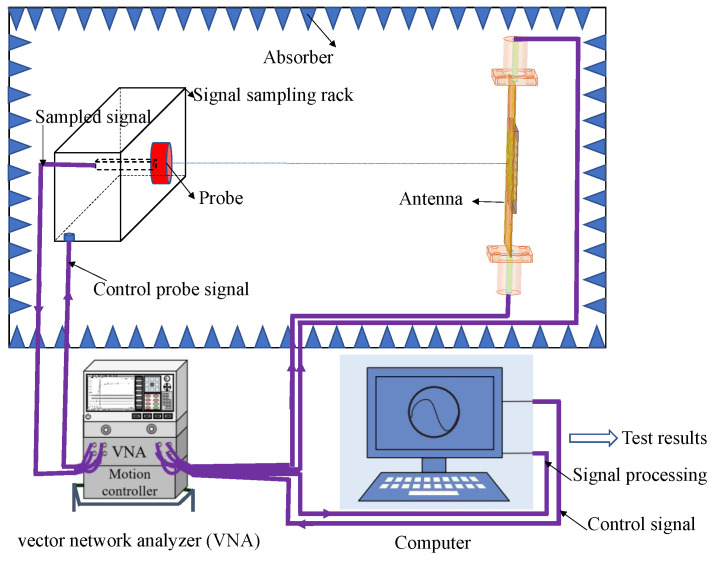
Schematic diagram of antenna testing environment.

**Figure 12 sensors-24-01184-f012:**
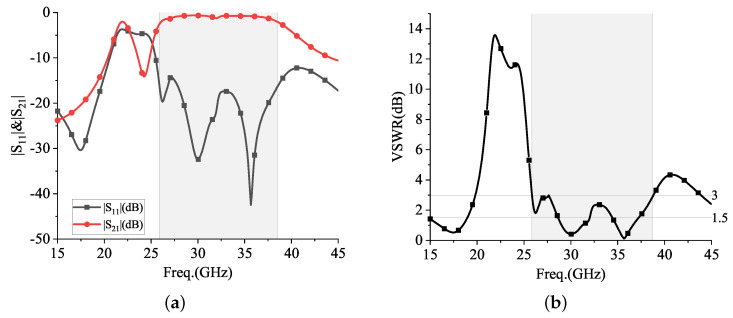
|S|-parameters and VSWR: (**a**) |S|-parameters; (**b**) VSWR.

**Figure 13 sensors-24-01184-f013:**
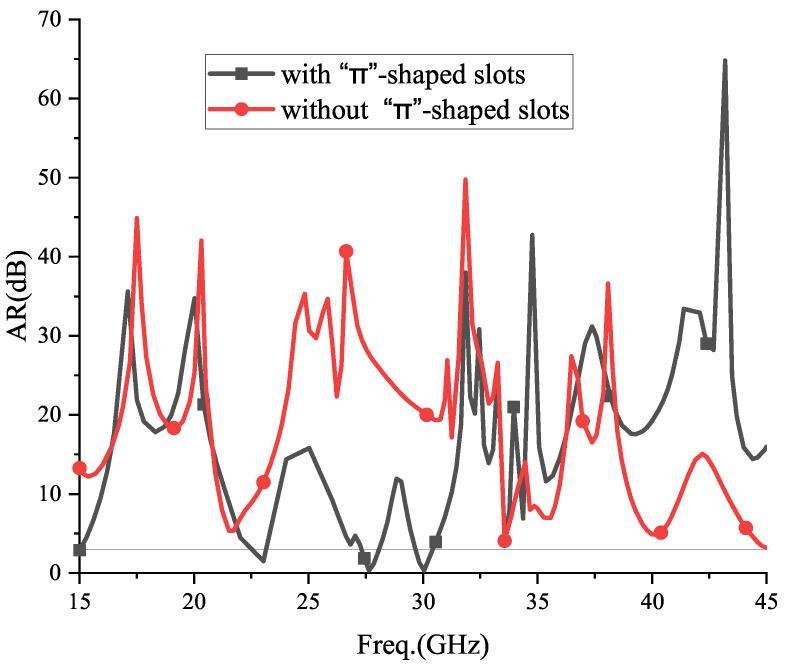
For an AR performance, it is compared with the proposed antenna while there is no “Π”-shaped slot.

**Figure 14 sensors-24-01184-f014:**
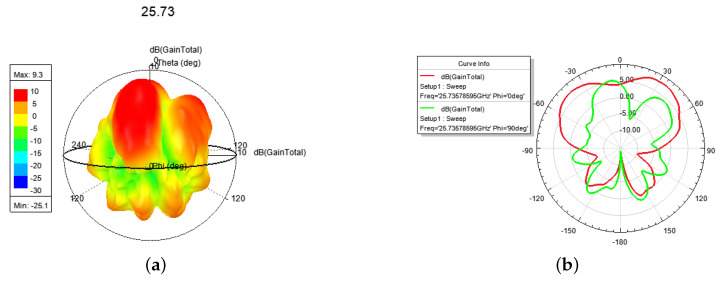
The 3-D and E-, H-plane directional pattern at 25.73 GHz: (**a**) 3-D, (**b**) E-, H-plane.

**Figure 15 sensors-24-01184-f015:**
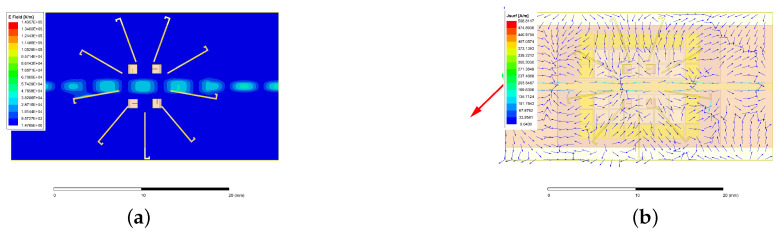
Electric field and current at 28.0 GHz: (**a**) electric field, (**b**) current.

**Figure 16 sensors-24-01184-f016:**
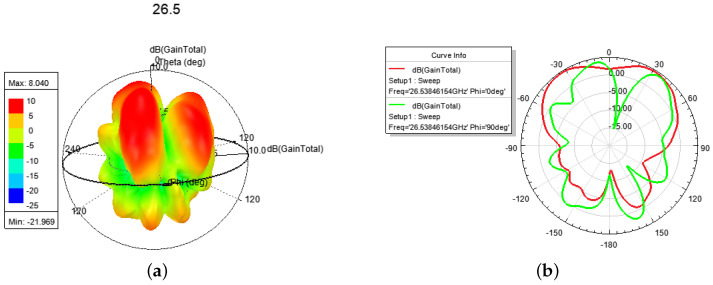
The 3-D and E-, H-plane directional pattern at 26.5 GHz, 27.5 GHz, 28.0 GHz: (**a**,**c**,**e**) 3-D, (**b**,**d**,**f**) E-, H-plane.

**Figure 17 sensors-24-01184-f017:**
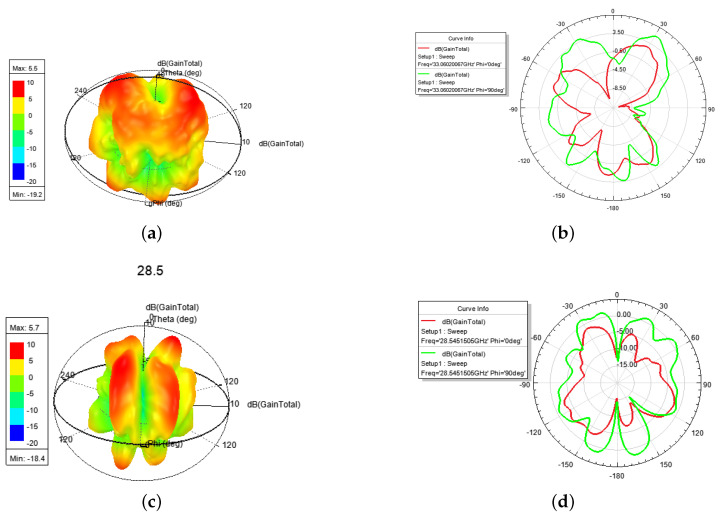
The 3-D and E-, H-plane directional pattern at 28.5 GHz: (**a**) 3-D, (**b**) E-, H-plane.

**Table 1 sensors-24-01184-t001:** Symbols and dimensions of elements (units: mm).

Parameters	$l_{s1-1}$	$l_{s1-2}$	$l_{s1-3}$	$w_{s1}$	$l_{s2-1}$	$l_{s2-2}$
Values	5.40	0.50	0.30	0.15	0.90	1.05
Parameters	$l_{s2-3}$	$w_{s2}$	$x_{s1}$	$x_{s2}$	$y_{s1}$	$y_{s2}$
Values	1.0	1.0	3.0	1.25	0.9	1.25

**Table 2 sensors-24-01184-t002:** Symbols and dimensions of SIGW (units: mm).

Parameters	$h_{1}$	$h_{2}$	$l_{\mathrm{MSL}}$	$w_{\mathrm{MSL}}$	$l_{\mathrm{SIGW}}$	$w_{\mathrm{SIGW}}$
Values	0.254	0.813	6	0.76	15.76	1.26

**Table 3 sensors-24-01184-t003:** Comparisons with the presented works in references.

Ref.	Tech.	$f_{0}$ (GHz)	FBW (%)	Gain (dBi)	Modes	Size (x,y,z) · $\lambda_{0}^{3}$
[5]	Pat.	2.4	2.92	–	CP–OAM	>(0.6,0.6,0.01)
[8]	Pat.-arr.	10.7	2.80	6.05	OAM	>(2.5,2.5,0.07)
[9]	Pat.-arr.	300	6.67	2	OAM	(7.0,7.0,0.05)
[11]	Meta-sur.	83.5	5.99	–	OAM	>(8.3,8.3,–)
[13]	Meta-sur.	10	10	13.9	LP, OAM	(10.0,10.0,2.5)
[14]	Meta-sur.	59	3.39	14.6	OAM	(–,–,–)
[18]	GW	5.5	7.27	12.2	LP	(3.5,0.7,0.28)
[7]	SIW	13	15.38	13	CP–OAM	(4.8,5.2,1.1)
[17]	SIW	60	26.1	16	CP–OAM	(1.38,1.38,0.15)
[19]	SIW	31	3.23	>11.4	CP–OAM	(8.2,5.4,0.16)
[22]	SIGW	60	35	5.1	CP	>(1.4,2.0,0.2)
[24]	SIGW	25	8.4	8.5	LP	(1.5,1.58,0.8)
Pro.	SIGW	28	35.7	8.04	L/CHP, CP–OAM	(1.5,1.4,0.2)

## Data Availability

Data available on request due to restrictions.

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
