# Peer review of "A Millimeter-Wave Broadband Multi-Mode Substrate-Integrated Gap Waveguide Traveling-Wave Antenna with Orbit Angular Momentum"

_sensors, 2024, doi:10.3390/s24041184_

Round 1

Reviewer 1 Report

Comments and Suggestions for Authors

1. In what ways does the antenna leverage Orbit Angular Momentum (OAM) for 5G communication, and how does this contribute to its functionality? Could the authors provide a more detailed explanation of the role of OAM in the proposed design?

2. The literature review should be expanded to include a more comprehensive overview of existing MM-wave antennas, especially those incorporating Orbit Angular Momentum (OAM), would provide a stronger foundation for the study.

3. A comparative analysis with existing MM-wave antennas, especially those utilizing OAM, would strengthen the paper's contribution by highlighting where the proposed antenna excels or differs.

4. While simulations are mentioned, there is a lack of detail regarding the simulation setup and parameters. Additionally, if experimental validation is conducted, it should be explicitly discussed, including the setup, conditions, and results.

5. The authors should explain how does the proposed antenna compare with existing models in terms of performance, bandwidth, and efficiency? Are there any notable advantages or limitations that distinguish it from current state-of-the-art designs?

6. The authors should elaborate on the implications and advantages of employing multi-mode operation in the proposed antenna? How does this feature contribute to the antenna's overall performance, especially in the context of 5G communication?

7. How well does the proposed antenna align with current or emerging 5G communication standards? Are there any specific features that make it particularly suited for the evolving requirements of 5G networks?

Comments on the Quality of English Language

Minor editing of English language required.

Author Response

Reply to the Review Report (Reviewer 1)

We gratefully thank the editor and all reviewers for their time spent making their constructive remarks and useful suggestions, which have significantly raised the quality of the manuscript and have enabled us to improve the manuscript. Each suggested revision and comment, brought forward by the reviewers was accurately incorporated and considered. Below the comments of the reviewers are responses point by point the revisions are indicated.

  1. Comment: In what ways does the antenna leverage Orbit Angular Momentum (OAM) for 5G communication, and how does this contribute to its functionality? Could the authors provide a more detailed explanation of the role of OAM in the proposed design?
  2. Reply: We appreciate your valuable comment. For the problem, first, we reply to the sentence “In what ways does the antenna leverage Orbit Angular Momentum (OAM) for 5G communication, and how does this A contribute to its functionality?” by using Fig.1 shown in the current version. As described in lines 117~128 in the current version, The OAM antenna utilizes the OAM mode through multiplexing, encoding information on the mode order l. Therefore, the more modes there are, the greater the information capacity.

For the comment: “ Could the authors provide a more detailed explanation of the role of OAM in the proposed design?” we reiterate in the abstract that this article introduces traditional slot antennas to OAM and clarifies OAM as the main role of this paper. This is also further explained in section 2. The role of OAM is elaborated more in Figures 15, 16, and 17, as well as Table 3, in the current version which provides a more detailed description.

These part modifications have been marked by using yellow color in the mdpi_modification_ reviewer1 revision.

  1. Comment: The literature review should be expanded to include a more comprehensive overview of existing MM-wave antennas, especially those incorporating Orbit Angular Momentum (OAM), would provide a stronger foundation for the study.
  2. Reply: We appreciate your valuable suggestions. We did three works to enhance the Introduction section: 1. Considering the dimensions of the waveguide and radiation slot in this article can be designed according to the resonant frequency corresponding to the wavelength, in this revision, the title has been removed from “5G”; 2. In order to fully reflect the significance and development trend of OAM research, in this revision, we have replaced the references [2]-[4] in the original manuscript with the current versions [2]-[4]. According to references [2]-[4], it is not difficult to see that the antenna carrying OAM has always been a key research topic in 5G, and even the Chinese expert group has included it in the development plan for 6G [3]; 3. In the introduction, the current research status of OAM antenna design has been classified, corrected, and supplemented according to reference [4]. The corrections are as follows: 1) To more fully describe the research status of OAM in millimeter waves in recent years, the references [7], [11], [15], [19] in the original manuscript have been replaced with the current references [7], [11], [17], [19]. 2)Simplified the original manuscript's description of reference [13]. For 1), Additional information: The current manuscript focuses on research topics [7], [11], [14], and [19].

[7] Based on the open resonant cavity theory and circular array theory, a high-order Laguerre-Gaussian OAM mode (i.e. HE21) with a phase difference of 90° is generated using eight aperture-coupled patch antenna elements [7].

[11] To increase the communication range, a lens-integrated metamaterial structure (MMS) was proposed to generate two OAM modes with a high-directivity broad-bandwidth characteristic[11].

[14] A millimeter-wave ultralow-reflectivity SPP OAM was proposed. However, SPP systems usually can only generate a single OAM beam.

[19] A millimeter wave 2 × 2 broadband CP-OAM antenna array is proposed, whose elements consist of two layers of metasurface patch layers. The antenna utilizes a sequential rotation feeding method to simultaneously generate dual OAM modes. The results indicate that the impedance bandwidth (S11<10 dB) and 3 dB axis ratio (AR) are 32.3 % (from 26 to 36 GHz).

These part modifications have been marked in the section “Introduction” by using yellow color in mdpi_modification_ reviewer1 revision.

  1. Comment: A comparative analysis with existing MM-wave antennas, especially those utilizing OAM, would strengthen the paper's contribution by highlighting where the proposed antenna excels or differs.
  2. Reply: We appreciate your valuable suggestions. For this issue, first of all, considering the "existing MM-wave antennas, especially those utilizing OAM", in the literature comparison Table 3, we replaced the original references [12], [19] with the current references [11], [19]. Secondly, we also added the current versions of the references [7], [14], [17] to the comparison. For a detailed comparison, please see lines 304-315.

These part modifications have been marked in the section “Introduction” by using yellow color in the mdpi_modification_reviewer1 revision.

  1. Comment: While simulations are mentioned, there is a lack of detail regarding the simulation setup and parameters. Additionally, if experimental validation is conducted, it should be explicitly discussed, including the setup, conditions, and results.
  2. Reply: We appreciate your valuable suggestions. For this issue, first of all, According to your opinion, as shown in Figure 10, we presented a detailed description of an operation interface for an HFFS.2020 simulated antenna, as described in lines 238-254 in the article.

Furthermore, experimental validation is conducted, we present a block diagram of the testing antenna, as shown in Figure 11, with a detailed description, please see lines 255-265

These part modifications have been marked by using yellow color in the mdpi_modification_ reviewer1 revision.

  1. Comment: The authors should explain how does the proposed antenna compare with existing models in terms of performance, bandwidth, and efficiency? Are there any notable advantages or limitations that distinguish it from current state-of-the-art designs?
  2. Reply: We appreciate your valuable suggestions. firstly, in the literature comparison Table 3, we replaced the original [12], [19] with the current versions [11], [19]. Secondly, we also added the current versions of the literature [7], [14], [17] to the comparison. Finally, for a detailed comparison, please see lines 304-315. The significant feature is that substrate-integrated gap waveguide (SIGW) introduces phase delay, resulting in coupling to the radiate units with a phase jump. Nine “L”-shaped slot radiate units are cut in a circular order at a certain angle on the SIGW to generate spin angular momentum (SAM) and orbit angular momentum (OAM). This is different from the current advanced design.

These part modifications have been marked by using yellow color in the mdpi_modification_ reviewer1 revision.

  1. Comment: The authors should elaborate on the implications and advantages of employing multi-mode operation in the proposed antenna? How does this feature contribute to the antenna's overall performance, especially in the context of 5G communication?
  2. Reply: The current version has already elaborated on the 5G communication environment and requirements in detail: 5G is facing the needs that the broadband, multiple band, high-capacity, and high-data-transmission-rate requirements. Orbit angular momentum (OAM) is considered a new dimension for improving the channel capacity in recent years, especially, its multiple orthogonal modal features. We propose this OAM antenna has a millimeter-wave broadband multi-mode characteristics. Therefore, this work has significant value and significance for 5G communication.

These part modifications have been marked by using yellow color in the mdpi_modification_ reviewer1 revision.

  1. Comment: How well does the proposed antenna align with current or emerging 5G communication standards? Are there any specific features that make it particularly suited for the evolving requirements of 5G networks?
  2. Reply: As described in detail in lines 227-233 of the current version: Based on the 5G millimeter wave bands, which include 21.4~22.0GHz,24.25~27.5GHz, 27.9 ~28.2 GHz,31.0~31.3 GHz, and 38.0~ 39.5 GHz [25,28], announced by the Wires Radiocommunication Conference-19 (WRC-19), the SIGW operates at 21.0~40.0GHz. The SIGW’s insertion loss is equal to 0.8 dB and RL is > 20 dB at 25.5~34.5GHz. In addition, its radiation structure is based on a generalized slot antenna and does not have high requirements for 5G systems. So, this job is suitable for the development needs of 5G networks.

These part modifications have been marked by using yellow color in the mdpi_modification_ reviewer1 revision.

Reviewer 2 Report

Comments and Suggestions for Authors

The abstract presents a novel approach to enhance channel capacity by incorporating Orbit Angular Momentum (OAM) in a 5G mm-wave broadband multi-mode waveguide traveling-wave antenna. The design utilizes the characteristics of Integrated Substrate Gap Waveguide (ISGW) to introduce phase delay, resulting in coupling to radiate units with a phase jump. The antenna includes "L"-shaped slot radiate units arranged in a circular order to generate Spin Angular Momentum (SAM) and OAM. Additional "Π"-shaped slot radiate units are strategically placed in a circular array to generate more OAM modes and match the antenna.

Simulation results indicate that the proposed antenna operates at 28 GHz, with a 3-dB fraction bandwidth (FBW) of 35.7%, ranging from 25.50 to 35.85 GHz. The Voltage Standing Wave Ratio (VSWR) is below 1.5 dB from 28.60 to 32.0 GHz. The antenna exhibits various modes, including a linear polarization (LP) mode with a gain of 9.3 dBi at 34.0 to 37.2 GHz, a l = 2 SAM-OAM (circular polarization OAM) mode with 8.04 dBi at 25.90 to 28.08 GHz, a l= 1 and l= 2 hybrid OAM mode with 5.7 dBi at 28.08 to 29.67 GHz, a SAM (left/right circular polarization) mode with 4.6 dBi at 29.67 to 30.41 GHz, and a LP mode at 30.41 to 35.85 GHz.

Additionally, the waveguide transmits bandwidth ranging from 26.10 to 38.46 GHz, maintaining quasi-TEM mode and an energy transmission loss (S21) of ≤ 2 dB.

Overall, the abstract provides a clear and comprehensive overview of the proposed antenna design, its operating characteristics, and performance metrics. The experimental results appear promising, showcasing the potential of the proposed antenna for 5G applications.

1. these new papers can improve the introduction:

    •           1.  DOI: 10.1109/MMM.2023.3269619

            2.DOI:10.1038/s41598-023-38247-x

            3. DOI: 10.1109/LAWP.2023.3338511

             4.  DOI: 10.3390/s23052702

    • 2. Please show field distribution and explain the Mushroom-type EBG application more. 
    • 3.Describe the effect of mushroom patch dimensions on s-parameters.
    • 4. the explanation of the Figure 6. Compared with MSL in S-parameters is not enough and confusing. please describe clearly.
    •  
    •  

    •  

Author Response

Reply to the Review Report (Reviewer 2)

We gratefully thank the editor and all reviewers for their time spent making their constructive remarks and useful suggestions, which have significantly raised the quality of the manuscript and have enabled us to improve the manuscript. Each suggested revision and comment, brought forward by the reviewers was accurately incorporated and considered. Below the comments of the reviewers are responses point by point the revisions are indicated.

  1. Comment: 1. these new papers can improve the introduction:

1). DOI:10.1109/MMM.2023.3269619

2). DO1:10.1038/s41598-023-38247-X3

3). DOI:10.1109/LAWP.2023.3338511

4). DOI: 10.3390/s23052702.

  1. Reply: We appreciate your valuable suggestions. We did three works to enhance the Introduction section: 1. Considering the dimensions of the waveguide and radiation slot in this article can be designed according to the resonant frequency corresponding to the wavelength, in this revision, the title has been removed from “5G”; 2. In order to fully reflect the significance and development trend of OAM research, in this revision, we have replaced the references [2]-[4] in the original manuscript with the current versions [2]-[4]. According to references [2]-[4], it is not difficult to see that the antenna carrying OAM has always been a key research topic in 5G, and even the Chinese expert group has included it in the development plan for 6G [3]; 3. In the introduction, the current research status of OAM antenna design has been classified, corrected, and supplemented according to reference [4]. The corrections are as follows: 1) To more fully describe the research status of OAM in millimeter waves in recent years, the references [7], [11], [15], [19] in the original manuscript have been replaced with the current references [7], [11], [17], [19]. 2)Simplified the original manuscript's description of reference [13]. For 1), Additional information: The current manuscript focuses on research topics [7], [11], [14], and [19].

[7] Based on the open resonant cavity theory and circular array theory, a high-order Laguerre-Gaussian OAM mode (i.e. HE21) with a phase difference of 90° is generated using eight aperture-coupled patch antenna elements [7].

[11] To increase the communication range, a lens-integrated metamaterial structure (MMS) was proposed to generate two OAM modes with a high-directivity broad-bandwidth characteristic[11].

[14] A millimeter-wave ultralow-reflectivity SPP OAM was proposed. However, SPP systems usually can only generate a single OAM beam.

[19] A millimeter wave 2 × 2 broadband CP-OAM antenna array is proposed, whose elements consist of two layers of metasurface patch layers. The antenna utilizes a sequential rotation feeding method to simultaneously generate dual OAM modes. The results indicate that the impedance bandwidth (S11<10 dB) and 3 dB axis ratio (AR) are 32.3 % (from 26 to 36 GHz).

These part modifications have been marked in the section “Introduction” by using green color in the mdpi_modification_reviewer2 revision.

  1. Comment: Please show field distribution and explain the Mushroom-type EBG application more.
  2. Reply: The field of the waveguide is shown in Figure 7, and its analysis is described in lines 214-225 Generation of EGB bandgap. The OAM mode exhibits a divergent state, resulting in low gain and more severe in high-order OAM modes. However, EBG improves the dispersion problem of traditional microstrip lines in millimeter waves, allowing only quasi-TEM propagation within the band and reducing energy loss. Therefore, compared to using microstrip lines to design the feeding structure of millimeter-wave OAM antennas, SIGW with EBG structure as the feeding structure of millimeter-wave OAM antennas has certain improvements in improving antenna gain.

In addition, the field of the waveguide traveling wave antenna with slits is shown in Figure 15. The analysis description can be found in lines 286-289.

These part modifications have been marked in the section “Introduction” by using green color in the mdpi_modification_reviewer2 revision.

  1. Comment: Describe the effect of mushroom patch dimensions on s-parameters.
  2. Reply: Here, we focus on analyzing the impact of microstrip lines combined with mushroom-shaped structures on the S parameter, as shown in Figures 8 and 9. The analysis is described in lines 214-222

These part modifications have been marked in the section “Introduction” by using green color in the mdpi_modification_reviewer2 revision.

  1. Comment: the explanation of the Figure 6. Compared with MSL in S-parameters are not enough and confusing. please describe clearly.
  2. Reply: Structurally speaking, SIGW is a waveguide made up of microstrip lines packed by a layer of periodic electromagnetic bandgap (EBG) structure units, which is a type of metamaterial known as a perfect magnetic conductor (PMC). In this article, we focus on analyzing the role of EBG structures in providing frequency selectivity and solving dispersion problems in waveguides, with a detailed description in lines 226-236 of the article.

These part modifications have been marked in the section “Introduction” by using green color in the mdpi_modification_reviewer2 revision.

Round 2

Reviewer 1 Report

Comments and Suggestions for Authors

The authors have incorporated all the comments appropriately in the revised manuscript.

Reviewer 2 Report

Comments and Suggestions for Authors

I accept this manuscript to be published as paper.